

# Antimicrobial and anti-endotoxin activity of N-acetylcysteine, calcium hydroxide and their combination against *Enterococcus faecalis, Escherichia coli* and lipopolysaccharides

Rayana Duarte Khoury[1], Amjad Abu Hasna[1,2], Carolina Fedel Gagliardi[1], Renata Marques de Melo Marinho[3], Cláudio Antonio Talge Carvalho[1], Eduardo Bresciani[4] and Marcia Carneiro Valera[1]

[1] Department of Restorative Dentistry, Endodontics Division, Institute of Science and Technology, Campus of São José dos Campos, São Paulo State University (ICT-Unesp), São José dos Campos, São Paulo, Brazil
[2] School of Dentistry, Universidad Espíritu Santo, Samborondón, Ecuador
[3] Department of Dental Materials and Prosthodontics, Institute of Science and Technology, Campus of São José dos Campos, São Paulo State University (ICT-UNESP), São José dos Campos, São Paulo, Brazil
[4] Department of Restorative Dentistry, Institute of Science and Technology, Campus of São José dos Campos, São Paulo State University (ICT-Unesp), São José dos Campos, São Paulo, Brazil

Corresponding author
Amjad Abu Hasna,
d.d.s.amjad@gmail.com

## ABSTRACT

**Background:** The management of endodontic infections is a complex challenge, mainly due to the involvement of diverse microorganisms and their by-products. This study aimed to evaluate the efficacy of N-acetylcysteine (NAC), calcium hydroxide ($Ca(OH)_2$), and their combined application as intracanal medications in combating *Enterococcus faecalis, Escherichia coli*, and lipopolysaccharides (LPS) from *E. coli*.

**Methods:** A total of 60 single-rooted human teeth were carefully selected and divided into six groups. These tooth canals were deliberately exposed to *E. faecalis* (ATCC 29212) and *E. coli* (ATCC 25922) to induce biofilm formation. Subsequently, the specimens were treated with NAC, $Ca(OH)_2$, or a combination of both substances. Three samples of the root canals were collected at three moments: the first sample (S1) was to confirm the initial contamination, the second sample (S2) was immediately post-instrumentation, and the third sample (S3) was collected after the use of the intracanal medication. The antimicrobial efficacy of these intracanal medications was assessed by enumerating colony-forming units per milliliter (CFU/mL). In addition to this, the kinetic chromogenic Limulus Amebocyte Lysate (LAL) assay by Lonza was used to quantify LPS from *E. coli*. Data tested for normality; then, Kruskal-Wallis and Friedman tests were used, and Dunn's for multiple comparisons.

**Results:** The findings of this study showed significant reductions in the microbial load of *E. faecalis* and *E. coli* by S3. Notably, there were no statistically significant differences among the treatment groups concerning these microorganisms. However, it was observed that only the combination of NAC and $Ca(OH)_2$ led to a noteworthy decrease in the quantity of *E. coli's* LPS after 7-days, demonstrating a statistically significant difference from the other treatment groups. NAC + $Ca(OH)_2$

combination, applied for a duration of 7-days, proved to be more suitable in reducing the presence of *E. faecalis*, *E. coli*, and LPS from *E. coli* within the context of endodontic infections.

## INTRODUCTION

The endodontic infection has a complex nature because of the involvement of various Gram-positive and Gram-negative microorganisms, endotoxins (lipopolysaccharides), that can initiate an organic response, resulting in the release of matrix metalloproteinases (MMPs) and cytokines (*Aw, 2016*; *Gomes & Herrera, 2018*; *Carvalho et al., 2020*). Thus, controlling this infection requires the use of antimicrobial agents with a wide range of efficacy that act against microorganisms and their byproducts (*Abu Hasna et al., 2020a*, *2020b*; *de Oliveira et al., 2022*; *Domingues et al., 2023*). Additionally, these agents play an important role in favoring the periapical healing process (*de Oliveira et al., 2024*).

*Enterococcus faecalis* is a Gram-positive bacterium; it is involved in both primary and secondary endodontic infections (*Pourhajibagher, Ghorbanzadeh & Bahador, 2017*), being one of the most prevalent bacteria (*Machado et al., 2020*). It can survive and regrowth days after the treatment (*Zandi et al., 2016*). Even more, it is not eliminated during the root canal treatment (*Cavalli et al., 2017*) because of its capacity to form biofilms and survive in alkaline pH (*Alghamdi & Shakir, 2020*), and in a recent report, was found to tolerates acidic environments (pH 2.9–4.2) (*Mubarak & Soraya, 2018*).

*Escherichia coli* is a Gram-negative bacterium involved in the endodontic infection and can be eliminated by endodontic treatment (*Valera et al., 2016*); however, it is studied due to the resistance of its lipopolysaccharide (LPS). LPS of Gram-negative bacteria could be liberated after bacteria death from its cell wall (*Stashenko, Teles & D'Souza, 1998*). LPS is present in high concentrations in root canals of symptomatic teeth has a positive correlation with the presence of endodontic signs and symptoms (*Cardoso et al., 2015*). It induces osteoclastogenic signaling, which culminates in bone resorption (*Ribeiro-Santos et al., 2019*). In addition, it is not detoxified completely by endodontic treatment (*Cavalli et al., 2017*).

Calcium hydroxide (Ca(OH)$_2$) is a widely used intracanal medication that detoxify the existed endotoxins in the root canal system (*Oliveira et al., 2005*; *Maekawa et al., 2011*, *2013*), eliminates *E. coli* (*Valera et al., 2016*); however, it is efficacy against *E. faecalis* is controversial in the literature (*Maekawa et al., 2013*; *Abu Hasna et al., 2020a*). Thus, increased necessity for combined intracanal medication is encouraged to obtain higher success (*Maekawa et al., 2013*).

N-acetyl cysteine (NAC) was reported primarily in endodontics as an effective chemo-protectant sealer (*Paranjpe et al., 2008*), as a possible anti-inflammatory for post-operative pain (*Ehsani et al., 2012*) and lastly as an intracanal medication due to its

efficacy against variety of endodontic pathogens (*Quah et al., 2012*; *Moon et al., 2016*) including *E. faecalis* (*Abu Hasna et al., 2020a*), and other bacteria species resistant to $Ca(OH)_2$ (*Martinho et al., 2023*). However, a basic study concluded that mixing $Ca(OH)_2$ with NAC is not recommended against *E. faecalis* (*Adl et al., 2022*). However, this study was motivated to understand the mechanism of action against different bacteria and their endotoxins.

Since there is not an ideal intracanal medication that acts on all the microbiota and its by-products in the root canal and is also effective in controlling the periapical inflammatory process the aim of this study is to evaluate the antimicrobial and anti-endotoxin activity of NAC, $Ca(OH)_2$ and their combined effect against *E. faecalis, E. coli*, and LPS of *E. coli*. The null hypothesis was that these medications have no antimicrobial action against the tested bacteria and have no anti-endotoxin action against LPS of *E. coli*.

## MATERIALS AND METHODS

### Preparation of specimens

The current research was conducted with the approval of the Human Ethics Committee at São Paulo State University, Brazil (approval number 4.002.491). Free and informed consent forms were obtained from all donors. A selection was made of sixty human teeth with single roots that exhibited dimensional and morphological similarities. After crosscutting the crowns using a carborundum disc, the roots were standardized to a length of 16 ± 0.5 mm. Each of the specimens underwent instrumentation up to K-file #30 (Dentsply Ind. Com. Ltda, Petrópolis, RJ, Brazil), utilizing 3 mL of a 1% NaOCl solution as an irrigant. The canals were filled with 17% trisodium ethylenediaminetetraacetic acid (EDTA) solution (Inodon, Porto Alegre, RS, Brazil) for a duration of 3 min, followed by rinsing with 10 mL of sterile saline solution. To seal the apical portions of the teeth, light-cured composite resin (Z-100, 3M, Sumaré, SP, Brazil) was applied, while the outer root surfaces were coated with two layers of nail polish, except for the cervical opening area. The specimens were then randomly allocated into six groups (Table 1), each containing ten specimens. These were affixed within 24-well cell culture plates using chemically activated acrylic resin, as outlined by *Matos et al. (2019)*. All materials utilized in this study were subjected to sterilization through gamma radiation utilizing cobalt 60 (20 KGy for a duration of 6 h), as detailed by *Csako et al. (1983)*.

### Contamination and preparation of specimens

To begin, a suspension of *E. coli* (ATCC 25922) containing $10^6$ cells/mL was prepared. Subsequently, 5 µL of this suspension was introduced into each root canal, followed by the addition of 10 µL of brain heart infusion (BHI) broth (Himedia Laboratories, Mumbai, India). Following a span of 7 days, another suspension was formulated, this time consisting of $10^6$ cells/mL of *E. faecalis* (ATCC 29212). Once again, 5 µL of this suspension was inoculated into each root canal, succeeded by an application of 10 µL of BHI broth. Throughout the entire incubation period, all specimens were consistently covered with a sterile cotton pellet soaked in the culture medium. These specimens were securely stored

**Table 1 The experimental groups.**

| Experimental group | Intracanal medication | Application period |
| --- | --- | --- |
| Ca(OH)$_2$ 7 d | Calcium hydroxide | Seven days |
| NAC 7 d | N-acetylcysteine | Seven days |
| Ca(OH)$_2$ + NAC 7 d | Calcium hydroxide and N-acetylcysteine | Seven days |
| Ca(OH)$_2$ 14 d | Calcium hydroxide | Fourteen days |
| NAC 14 d | N-acetylcysteine | Fourteen days |
| Ca(OH)$_2$ + NAC 14 d | Calcium hydroxide and N-acetylcysteine | Fourteen days |

within an incubator set at a temperature of $37 \pm 1$ °C under controlled relative humidity conditions. Over the course of the incubation, BHI broth was replenished within the root canals every 2 days, spanning a total of 28 days for the *E. coli* incubations and 21 days for the *E. faecalis* incubations.

The root canals were instrumented using the RECIPROC system file R40 (VDW–Germany), which was coupled to an electric motor (VDW) to facilitate a reciprocating movement. This instrumentation process encompassed the entire length of the canals and was accompanied by the irrigation of 5 mL of sterile saline solution for each one-third segment, amounting to a total irrigation volume of 15 mL.

## Sample collection

Three distinct samples were collected utilizing sterile paper points. The initial sample, labeled as S1, was gathered using paper points of size #25 (Dentsply Maillefer, Ballaigues, Switzerland) with the purpose of confirming the presence of specimen contamination. Subsequently, the second sample, referred to as S2, was obtained immediately following the instrumentation procedure using paper points of size #40. Lastly, the third sample, denoted as S3, was collected after a period of 7 to 14 days following the application of intracanal medication.

All sample collections adhered to an identical protocol. Paper points were introduced into the root canal up to its working length and allowed to remain in place for a duration of 60 s. Following this, the paper points were then transferred to sterile microtubes containing 1,000 μL of sterile saline solution, as outlined by *Valera et al. (2010)*.

## Intracanal intracanal medication preparation

1) Ca(OH)$_2$ group: The Ca (OH)$_2$ powder obtained from (Biodinâmica Química e Farmacêutica LTDA, Paraná, Brazil) was combined with sterile saline solution in a 1:1 ratio (1 g of powder and 1mL of saline). This mixture was manipulated on a sterile glass plate using a spatula until it achieved a toothpaste-like consistency. Subsequently, the paste was introduced into the root canal using a lentulo instrument and completed with a K-file #30.

2) NAC group: The NAC powder sourced from (Sigma-Aldrich, St. Louis, MO, USA) was blended with saline in the same 1:1 proportion (1g of powder and 1mL of saline) as

outlined for the Ca(OH)$_2$ group. The resulting paste was inserted into the root canal using a K-file #30, replicating the procedure used for the Ca(OH)$_2$ group.

3) Ca(OH)$_2$ + NAC combined group: A mixture of 500 mg of Ca(OH)$_2$ powder and 500 mg of NAC powder was created and manipulated with saline in the same manner as the other groups (1:1 proportion, 1g of powder and 1mL of saline). This paste was inserted into the root canal using the established procedure.

All specimens were stored at 37 °C for 7/14 days. The medications were then removed with 10 mL of saline, and a new sample was collected with paper point #45 (S3).

### Culture procedure

To determine the antimicrobial activity, material collected through paper points was shaken and serial dilutions were made and 100 µL aliquots were seeded into duplicate petri dishes containing Enterococcosel agar (Himedia Laboratories, Mumbai, India) for *Enterococcus faecalis*, and MacConkey agar (Himedia Laboratories, Mumbai, India) for *Escherichia coli.*. Then, the plates were incubated at 37 °C for 48 h for later counting of colony-forming units/mL (CFU/mL).

### Quantification of endotoxins (LPSs): kinetic chromogenic LAL assay

The quantification of endotoxins was conducted using the kinetic chromogenic LAL assay provided by Lonza. In this essay, the LPS of *E. coli* served as the standard for reference. To ensure the accuracy of results, a positive control was included for each sample, involving a root canal sample intentionally contaminated with a known quantity of endotoxin. This step was crucial in evaluating the presence or absence of any interfering agents.

In the testing process, a 96-well apyrogenic plate was utilized. It contained the following components: 100 µL of apyrogenic water (serving as a reaction blank), five standard endotoxin solutions with concentrations ranging from 0.005 to 50 endotoxin units/mL, the root canal samples, and positive controls (each containing a known concentration of endotoxin, specifically 10 endotoxin units/mL). This comprehensive setup was replicated in four separate wells to ensure precision.

The plate was subjected to an incubation period of 10 min at a constant temperature of 37 ± 1 °C within a kinetic-QCL reader (Lonza, Walkerville, MI, USA), which was seamlessly connected to a microcomputer running the WinKQCL software (Lonza). Following this incubation, 100 µL of chromogenic reagent was added to each well. The kinetic test was initiated, during which the software meticulously tracked the absorbance at 405 nm for each well within the microplate. This data was then employed to automatically compute the log/log linear correlation between the reaction time of each standard solution and the corresponding concentration of endotoxin.

### Statistical analysis

Normality test was used after obtaining data. Kruskal-Wallis and Friedmann's tests were used to compare the obtained data and Dunn's for multiple comparison among the experimental groups.

**Table 2 Median and range of *E. coli* and CFU/mL count for all groups at baseline samples (S1), after instrumentation (S2), after intracanal medication (S3).**

| Groups | *E. coli* | | |
|---|---|---|---|
| | **S1** | **S2** | **S3** |
| Ca(OH)$_2$ 7 days | 17,391 (100–165*10$^3$) | 4,007 (0–10,400) | 0 (0–0) |
| | A-a | A-a | A-b |
| NAC 7 days | 13,110 (400–103*10$^3$) | 6,736 (0–45,200) | 0 (0–0) |
| | A-a | A-a | A-b |
| Ca(OH)$_2$ + NAC 7 days | 15,358 (3,780–86*10$^3$) | 471 (10–870) | 0 (0–0) |
| | A-a | AB-ab | A-b |
| Ca(OH)$_2$ 14 days | 504,600 (6,000–113*10$^4$) | 1,868 (0–5,800) | 1.1 (0–10) |
| | B-a | AB-ab | A-b |
| NAC 14 days | 261,600 (9,000–564*10$^4$) | 260 (0–600) | 0 (0–0) |
| | AB-a | AB-ab | A-b |
| Ca(OH)$_2$ + NAC 14 days | 398,200 (30*10$^3$–222*10$^4$) | 68 (0–230) | 0 (0–0) |
| | B-a | B-b | A-b |

**Note:**
Uppercase letters indicate the differences among groups. Lowercase letters indicate the differences among the samples of each group.

# RESULTS

## *E. coli*

All the experimental groups presented significant statistical difference between S1 and S3 in which all the medications were effective in reducing the microbial load. Among the experimental groups there was no statistical difference in S3 (Table 2)

## *E. faecalis*

Similar results were obtained, in which all the medications were effective against *E. faecalis* presenting a significant difference between S1 and S3; however, there was no statistical difference among the experimental groups in S3 (Table 3)

## LPS (endotoxin)

The biomechanical preparation was effective in detoxifying the LPS in all experimental groups and presented a significant difference between S2 and S1. All the experimental groups increased the LPS after using intracanal medications, except for the Ca(OH)$_2$+NAC 7 days group. In the groups Ca(OH)$_2$+NAC 7 days, NAC 14 days and Ca(OH)$_2$+NAC 14 days, statistical differences were observed between S1 and S3. On the other hand, the experimental groups Ca(OH)$_2$7 days, NAC 7 days and Ca(OH)$_2$14 days were statistical equal to S1 and S2 even reducing the LPS quantity (Table 4).

# DISCUSSION

Numerous studies have been conducted to better understand the behavior of intracanal medications and their efficacy against variety of micro-organisms present inside the root canal system (*Valera et al., 2010*, *2015*, *2016*; *Maekawa et al., 2011*; *Ooi et al., 2019*).

**Table 3 Median and range of *E. faecalis* and CFU/mL count for all groups at baseline samples (S1), after instrumentation (S2), and after intracanal medication (S3).**

| Groups | *E. faecalis* | | |
|---|---|---|---|
| | S1 | S2 | S3 |
| Ca(OH)$_2$ 7 days | 7,510 (4,000–27 * 10$^3$) | 2,997 (30–10,900) | 0 (0–0) |
| | A-a | A-a | A-b |
| NAC 7 days | 15,400 (3,000–30 * 10$^3$) | 1,320 (0–4,900) | 0 (0–0) |
| | AB-a | A-ab | A-b |
| Ca(OH)$_2$ + NAC 7 days | 5,050 (600–10 * 10$^3$) | 365 (70–900) | 0 (0–0) |
| | A-a | A-ab | A-b |
| Ca(OH)$_2$ 14 days | 181,900 (16,000–410 * 10$^3$) | 633 (30–3,000) | 0 (0–0) |
| | B-a | A-ab | A-b |
| NAC 14 days | 95,700 (48,000–180 * 10$^3$) | 949 (90–2,100) | 0 (0–0) |
| | AB-a | A-ab | A-b |
| Ca(OH)$_2$ + NAC 14 days | 110,500 (45 * 10$^3$–210 * 10$^3$) | 805 (60–2,200) | 0 (0–0) |
| | B-a | A-ab | A-b |

Note:
Uppercase letters indicate the differences among groups. Lowercase letters indicate the differences among the samples of each group.

**Table 4 Median and range of LPS counts for all groups at baseline samples (S1), after instrumentation (S2), and after intracanal medication (S3).**

| Groups | LPS | | |
|---|---|---|---|
| | S1 | S2 | S3 |
| Ca(OH)$_2$ 7 days | 114.29 (11.1–365) | 20.152 (1.92-47.8) | 28.947 (4–81.3) |
| | A-a | A-b | A-ab |
| NAC 7 days | 203.75 (13.4–599) | 24.82 (3.62–57.8) | 64.56 (13.8–170) |
| | A-a | A-b | A-ab |
| Ca(OH)$_2$ + NAC 7 days | 92.72 (11.7–253) | 22.49 (1.93–44.8) | 9.44 (4.73–21.2) |
| | A-a | A-ab | B-b |
| Ca(OH)$_2$ 14 days | 578.87 (8.66–933) | 5.59 (1.03–21.3) | 18.20 (2.31–41.8) |
| | B-a | B-b | A-ab |
| NAC 14 days | 466.72 (39.2–786) | 8.27 (1.51–24.8) | 22.09 (2.22–63.8) |
| | B-a | AB-b | A-b |
| Ca(OH)$_2$ + NAC 14 days | 724.7 (364–1,310) | 7.59 (3.38–14.6) | 62.99 (6.5–171) |
| | B-a | AB-b | A-b |

Note:
Uppercase letters indicate the differences among groups. Lowercase letters indicate the differences among the samples of each group.

The investigation into the combination of Ca(OH)$_2$ and N-acetylcysteine (NAC) is based on the potential synergistic effects these agents may offer in enhancing root canal disinfection. Although specific recommendations for the combined use of Ca(OH)$_2$ and NAC against *E. faecalis* are limited, we hypothesized that NAC, with its mucolytic and antioxidant properties, could potentially augment the antimicrobial efficacy of Ca(OH)$_2$

(*Olofsson, Hermansson & Elwing, 2003*). $Ca(OH)_2$ is well-established for its antimicrobial effects against various endodontic pathogens, although its efficacy is reduced against *E. faecalis* and *C. albicans* (*Carbajal Mejía, 2014*). Conversely, NAC has demonstrated efficacy in inhibiting biofilm formation and degrading proteinaceous and viscoelastic substances (*Olofsson, Hermansson & Elwing, 2003*). In addition, while $Ca(OH)_2$ alone did not elevate resolvin levels in apical periodontitis, NAC significantly increased RvE1 and RvD2 levels after 14 days (*Corazza et al., 2021*). Furthermore, combining $Ca(OH)_2$ with other agents, such as omeprazole, has been shown to enhance antimicrobial activity against *E. faecalis* and improve periapical lesion repair *in vivo* (*Wagner et al., 2011*; *Divakar et al., 2020*; *Anija et al., 2021*). These findings support the rationale of this study that the combination of $Ca(OH)_2$ and NAC might offer improved microbial reduction and enhanced therapeutic outcomes in endodontic treatment by exploiting their complementary mechanisms on *E. faecalis, E. coli*, and LPS of *E. coli*.

The results of the present study demonstrated that the combination of both intracanal medications was effective against *E. faecalis, E. coli*, and LPS from *E. coli* in root canals. According to the literature, only one study has evaluated the combined effect of NAC and $Ca(OH)_2$. In that study, the authors concluded that this combination is not recommended based on *in vitro* analysis of colony forming units of *E. faecalis* (*Adl et al., 2022*). Due to the lack of additional studies, it was not possible to compare these results with other literature.

Regarding the efficacy of intracanal medications on *E. coli*, it was found that all the intracanal medications were effective in reducing the microbial load without statistically significant differences among the groups. Specifically, $Ca(OH)_2$ demonstrated notable effectiveness against *E. coli*, consistent with the findings of *Valera et al. (2016)*, who reported that $Ca(OH)_2$ is effective when applied in the root canal for 14 days, and that this efficacy persists for 7 days after removal of the medication. This is in line with earlier results reported by *Valera et al. (2015)*, which also highlighted the sustained antimicrobial action of $Ca(OH)_2$ against *E. coli*. In the present study, $Ca(OH)_2$ showed comparable results to those found in the literature, with no significant differences observed between the 7- and 14-day application periods.

It is worth noting that a saline solution was used as the endodontic irrigant in this study, rather than more potent antimicrobial irrigants such as sodium hypochlorite or chlorohexidine (*Sonisha et al., 2024*; *Souza et al., 2024*). Saline solution is neutral and has minimal antimicrobial activity (*Tanvir et al., 2023*). Therefore, the use of saline would not influence the antimicrobial effectiveness of the tested intracanal medications (*Abu Hasna et al., 2020a*).

Examining the effects of N-acetylcysteine (NAC) over different time periods, the present study found that NAC was effective against *E. coli* whether applied in the root canal for 7 or 14 days, with no statistically significant difference between the two durations. *Marchese et al. (2003)* attributed this efficacy to NAC's ability to inhibit biofilm synthesis. However, another study by *Shen et al. (2020)* found that while NAC was effective in reducing *E. coli* biofilms, it did not achieve complete elimination of the biofilms. Similar results were reported by *El-Feky et al. (2009)* who concluded that NAC inhibits *E. coli* biofilm production and eradicates preformed mature biofilms. Notably, the optimal
duration for NAC to remain in the root canal was not specifically addressed in these studies, indicating a gap in the literature regarding the ideal application time.

Turning to the effects on LPS levels, the present study found that all experimental groups demonstrated similar results in reducing LPS levels after 7 and 14 days. The reduction of LPS levels achieved with $Ca(OH)_2$ after 7 and 14 days was statistically similar to the reduction observed after the biomechanical preparation. This finding is consistent with *Oliveira et al. (2005)* and *Marinho et al. (2018)*, that $Ca(OH)_2$ can effectively detoxify LPS in the root canal system. However, *Cavalli et al. (2017)* found that while endodontic treatment can detoxify LPS it does not completely remove it. Thus, although $Ca(OH)_2$ detoxifies LPS, it does not achieve complete removal, as also noted by *Oliveira et al. (2005)*.

Conversely, there are no studies in the literature evaluated the effect of NAC against LPS of *E.coli* in the root canal system. However, NAC has been reported to be effective in reducing LPS levels in LPS-induced lung injuries in rodents (*Mitsopoulos et al., 2008*).

The effect of $Ca(OH)_2$ against *E. faecalis* has been reported in various studies with inconsistent results. Some studies (*Valera et al., 2016*; *Ooi et al., 2019*) suggest that complete disinfection of *E. faecalis* can be achieved with $Ca(OH)_2$, and these findings are consistent with the present study, where $Ca(OH)_2$ was completely effective against *E. faecalis* after 7 and 14 days. However, another study by *Campanella et al. (2019)* indicated that while $Ca(OH)_2$ is effective against *E. faecalis*, it does not completely eliminate biofilms. Additionally, confocal microscopic and laboratory studies have reported some level of resistance of *E. faecalis* to $Ca(OH)_2$ (*Varshini et al., 2019*; *Moradi Eslami et al., 2019*; *Asnaashari et al., 2019*).

Studies have shown that NAC is effective against both planktonic and biofilm forms of *E. faecalis*, but it does not offer an advantage over $Ca(OH)_2$ (*Ulusoy et al., 2016*). This finding is consistent with the results of the present study where no statistical difference was observed between $Ca(OH)_2$ and NAC, regardless of the duration of their presence in the root canal system. However, *Quah et al. (2012)* reported that NAC completely eradicated *E. faecalis* biofilms and demonstrated a superior effect compared to $Ca(OH)_2$.

The role of NAC as an antimicrobial agent has been emphasized not only for its ability to reduce biofilm, prevent bacterial adhesion, and hinder the formation of these organized communities but also for its mucolytic effect (*Olofsson, Hermansson & Elwing, 2003*). In various *in-vitro* studies, NAC has proven effective in eliminating endodontic biofilms and reducing the prevalence of highly virulent bacterial species with significant pathogenic potential, such as *E. faecalis* (*Choi et al., 2018*; *Abdulrab et al., 2022*; *Adl et al., 2022*). Clinically, NAC has demonstrated antimicrobial activity against a broad range of bacterial species involved in primary endodontic infections, underscoring its potential for use in endodontic treatment (*Csako et al., 1983*). Additionally, the use of NAC as an intracanal medication has been associated with a significant increase in the levels of resolution, potent lipid mediators, anti-inflammatory, and immunomodulatory agents that contribute to the inflammation resolution pathway (*Corazza et al., 2021*).

The findings of this study highlight that both $Ca(OH)_2$ and NAC, as well as their combination, are effective as intracanal medications against *E. coli* and *E. faecalis*. All tested medications demonstrated efficacy in reducing LPS levels, although none were able

to eliminate LPS. Importantly, the effectiveness of these medications was consistent regardless of the duration of application in the root canal system. Despite the lack of a clear advantage of the combination therapy over the individual treatments in this study, the results offer valuable insights into their potential interactions and efficacy. Further research is necessary to fully elucidate the benefits and optimal application strategies of these treatments in clinical practice.

## CONCLUSIONS

- Both $Ca(OH)_2$ and NAC in addition to the combination of both, all are effective intracanal medication against *E. coli* and *E. faecalis*.
- All tested medication can reduce LPS levels but not eliminate LPS.
- Regardless of the application period in the root canal system, the intracanal medications were effective.

### Funding
This work was supported by The São Paulo Research Foundation (FAPESP) 2018/01703-9. The funders had no role in study design, data collection and analysis, decision to publish, or preparation of the manuscript.

### Grant Disclosures
The following grant information was disclosed by the authors:
São Paulo Research Foundation (FAPESP): 2018/01703-9.

### Competing Interests
Amjad Abu Hasna is an Academic Editor for PeerJ.

### Author Contributions
- Rayana Duarte Khoury conceived and designed the experiments, performed the experiments, analyzed the data, prepared figures and/or tables, authored or reviewed drafts of the article, and approved the final draft.
- Amjad Abu Hasna conceived and designed the experiments, performed the experiments, analyzed the data, prepared figures and/or tables, authored or reviewed drafts of the article, and approved the final draft.
- Carolina Fedel Gagliardi performed the experiments, authored or reviewed drafts of the article, and approved the final draft.
- Renata Marques de Melo Marinho conceived and designed the experiments, authored or reviewed drafts of the article, and approved the final draft.
- Cláudio Antonio Talge Carvalho conceived and designed the experiments, analyzed the data, authored or reviewed drafts of the article, and approved the final draft.
- Eduardo Bresciani conceived and designed the experiments, authored or reviewed drafts of the article, and approved the final draft.

- Marcia Carneiro Valera conceived and designed the experiments, performed the experiments, analyzed the data, authored or reviewed drafts of the article, and approved the final draft.

## Human Ethics

The following information was supplied relating to ethical approvals (*i.e.*, approving body and any reference numbers):

The current research was conducted with the approval of the Human Ethics Committee at São Paulo State University, Brazil.

## Data Availability

The raw measurements are available in the Supplemental File.

## Supplemental Information

Supplemental information for this article can be found online at http://dx.doi.org/10.7717/peerj.18331#supplemental-information.

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
