# Peer review of "Antimicrobial and anti-endotoxin activity of N-acetylcysteine, calcium hydroxide and their combination against *Enterococcus faecalis, Escherichia coli* and lipopolysaccharides"

_PeerJ, doi:10.7717/peerj.18331_

## Round 0.1 · original submission · Major Revisions

The authors should address all the comments from the reviewers. Also, there should be information in the introduction about what are the current drugs used and why should we used this combination of NAC and Calcium hydroxide. Also I feel that there should be a proper positive control in your experiment to show the efficacy/ inhibitory activity. The authors can respond to these in a timely manner or resubmit if they need more time.

Reviewer 1 ·

Basic reporting

The study aimed to evaluate the antimicrobial and anti-endotoxin activity of NCA and Calcium Hydroxide, as well as their combination, against E. faecalis and E. coli. I noticed some details that need to be improved or properly justified for the work to be published:

1) Throughout the manuscript, there are excessively old references (12 references from the last 5 years; 19 references from 5 to 10 years; and 13 references over 10 years old). There are several more recent articles (2023 and 2024) that could be explored to justify the research and highlight the novelty of the study.

2) In this regard, I highlight that the research gap and the novelty of the study are not explicitly stated in the introduction.

3) The self-citation index of the article is alarming (more than 33% of the cited articles are endogenous). Are there indeed few groups studying the object in question? Or does the object not attract the attention of other groups? Why?

Experimental design

Methods described with sufficient detail & information to replicate.

Validity of the findings

No comment

Reviewer 2 ·

Basic reporting

The aim of this work was to evaluate the efficaccy of N-acetylcysteine (NAC), Calcium hydroxide [Ca(OH)2], and their combined application as intracanal medications in combating Enterococcus faecalis, Escherichia coli, and lipopolysaccharides (LPS) from E. coli.

The manuscript is interesting and well-written. However, the authors should clarify the rationale for choosing sterile saline as the irrigant solution instead of an active irrigation solution. This clarification would strengthen the manuscript by providing a clear understanding of the decision-making process behind this choice.

Instead of E Faecalis, it should be E faecalis. Please correct it.

Experimental design

The work was well developed, and the methodology has been already published. However I would like to know if there was a reason for using only one type of culture medium to grow two different bacteria? How could the authors report the results of E faecalis and E coli if they used only one type of medium? Please clarify this point.

Validity of the findings

The raw data was provided. The findings were reported adequately. However it is necessary to clarify about E faecalis and E coli results.

Additional comments

How did the authors check the efficacy of the double contamination in the present work? Was there any MEV picture showing the intratubular contamination

---

## Round 0.2 · Major Revisions

Please go through the Reviewer comments and fix the language using professional help and rearrange the discussion as per suggestions.

Reviewer 2 ·

Basic reporting

The aim of this work was to evaluate the efficaccy of N-acetylcysteine (NAC), Calcium hydroxide [Ca(OH)2], and their combined application as intracanal medications in combating Enterococcus faecalis, Escherichia coli, and lipopolysaccharides (LPS) from E. coli.
The revised manuscript has become harder to read after incorporating the reviewers' feedback. Several sentences now start in a repetitive manner particularly in the Discussion section. Additionally, the rationale for combining CaOH2 and NAC is unconvincing, particularly since the authors themselves noted that this combination is not recommended for use against E. faecalis.

Experimental design

The work was well executed, and the methodology used has already been published. The authors incorporated the two types of media required to grow the two different bacteria

Validity of the findings

The raw data was provided. The findings were reported adequately.

Additional comments

My primary concern with the manuscript is that it has become difficult to read after the extensive revision, particularly in the last two paragraphs of the discussion section. These paragraphs seem out of place and disrupt the flow of the text. I recommend that the authors consider relocating these paragraphs to the middle of the discussion, where they would fit more naturally within the overall argument. This adjustment would help restore coherence to the discussion and make the manuscript easier to follow. The current ending of the discussion should then be revised to ensure a strong and logical conclusion to the section.

---

## Round 0.3 · accepted · Accept

The authors have addressed the reviewers comments satisfactorily. The manuscript is suitable for publication

Reviewer 2 ·

Basic reporting

The manuscript has improved significantly and is now much clearer and more concise than the previous versions, making it easier to follow and understand. The authors added to the text the reviewer’s recommendations, including the reasons for combining CaOH2 and NAC.

Experimental design

The work was well executed, and the methodology used has already been published. The authors incorporated the two types of media required to grow the two different bacteria. They also explained better the reasons for combining CaOH2 and NAC.

Validity of the findings

The raw data was provided. The findings were reported adequately

Additional comments

The manuscript is much better now.